# Hypertension-Related Knowledge, Attitudes, and Behaviors among Community-Dwellers at Risk for High Blood Pressure in Shanghai, China

**DOI:** 10.3390/ijerph17103683

**Published:** 2020-05-23

**Authors:** Dan Gong, Hong Yuan, Yiying Zhang, Huiqi Li, Donglan Zhang, Xing Liu, Mei Sun, Jun Lv, Chengyue Li

**Affiliations:** 1Department of Health Policy and Management, School of Public Health, Fudan University, P.O. Box 177, 130 Dong’ an Road, Shanghai 200032, China; 18211020057@fudan.edu.cn (D.G.); sunmei@fudan.edu.cn (M.S.); 2Research Institute of Health Development Strategies, Fudan University, Shanghai 200032, China; 3Jiading District Center for Disease Control and Prevention, Shanghai 201899, China; yhong11809@sohu.com (H.Y.); zyyzyyjd@163.com (Y.Z.); 4School of Public Health and Community Medicine, Institute of Medicine, University of Gothenburg, 40530 Gothenburg, Sweden; huiqi.li@amm.gu.se; 5Department of Health Policy and Management, College of Public Health, University of Georgia, Athens, GA 30602, USA; dzhang@uga.edu; 6Department of Epidemiology, School of Public Health, Fudan University, Shanghai 200032, China; liuxing@fudan.edu.cn

**Keywords:** population at risk for hypertension, knowledge-attitude-behavior, Eastern China, influencing factors

## Abstract

This study aimed to investigate the hypertension-related knowledge, attitudes, behaviors (KAB), and socio-demographic determinants among community-dwellers who were at risk for hypertension in Shanghai, China. A cross-sectional survey was conducted in a district of Shanghai in 2017 using multi-stage cluster sampling, including 611 participants who were at risk for hypertension. Questionnaires were used to investigate KAB regarding hypertension prevention. Multivariable logistic regression was performed to examine the relationship between socio-demographic factors and hypertension-related KAB. The results indicated that more than 75% of the study population had accurate knowledge, but only 48.4% knew the Recommended Daily Intake of salt for adults; over 80% formed health beliefs, while less than 50% were keeping a healthy diet, maintaining regular physical activity and/or bodyweight control. Better knowledge was found in the below 60 age group (*p* < 0.01) and the 60–69 age group (*p* = 0.03) than in the ≥70 age group. The behaviors in females (*p* < 0.01) were better than in males and were better in those covered by the Urban Employee Basic Medical Insurance (*p* = 0.01) than in those with the New Rural Cooperative Medical Insurance. In conclusion, although the rates of accurate knowledge and belief of hypertension prevention were high in the study population, the rates of maintaining healthy behaviors were relatively low. Socio-demographic factors had important influences on hypertension-related KAB. Further health education and intervention of hypertension prevention was needed to improve their level of KAB and reduce their risk for hypertension among the target groups.

## 1. Introduction

Hypertension is one of the most important public health problems in the world [1]. It is the major risk factor for myocardial infarction and stroke, which contributes to a tremendous disease burden and millions of deaths globally [2]. The prevalence of hypertension in adults increased from 18.8% to 25.2% between 2002 and 2012 in China [3]. It is estimated that there will be 352 million hypertensive patients in China by 2030 according to the current trend of increase in the prevalence [4], which will bring huge disease and economic burden to the families and the society.

The population at risk for hypertension is a group with prevalent risk factors or characteristics and hence a higher risk of hypertension than the normotensive people. Epidemiological studies have shown that prehypertensive people have doubled the risk of developing hypertension compared to individuals with normal blood pressure (BP) [5], and the morbidity and mortality of cardiovascular and cerebrovascular diseases are also higher [6]. In recent years, the increasing unhealthy lifestyles have led to a growth of the number of people at risk for hypertension [7]. The enormous amount of the high-risk subjects would become a huge pressure on individuals, families, and the society once they develop hypertension [8]. Therefore, it is of great importance to manage the high-risk individuals, as a key pathway to reduce the prevalence of hypertension.

To inform, educate and influence the general public’s knowledge, attitudes, and behaviors (KAB), particularly those who are at risk of developing hypertension is critical to prevent hypertension before it develops [9]. Several studies have implied that lifestyle interventions, such as salt restriction, increased exercise, and smoking cessation, can be effective in reducing the risk for hypertension [10].

The socio-demographic factors, including gender, age, educational level and income, have been considered as important determinants of the level of KAB [11]. For example, young people have better knowledge regarding hypertension than older people [12]; women have better overall health behaviors than men [13], and people with higher income tend to have more positive attitudes towards managing diseases [14]. Therefore, it is necessary to investigate which of these factors play important roles in the targeted high-risk population.

Several studies have investigated the KAB in the high-risk population, including the awareness of diagnostic criteria for hypertension, recognition of the benefit of hypertension prevention [15], and behaviors such as physical activity [16] and vegetable/fruit intake [17]. Some interventional studies analyzed the changes in KAB before and after the intervention. For example, after the intervention, the knowledge awareness rate increased and the smoking rate decreased [18]. However, few studies focused on the influencing factors of KAB in the high-risk population. To our knowledge, only limited studies have analyzed the influence of gender, age, educational level, and marital status on the KAB in Central China or rural areas of China [15,19], and no comparable studies have been conducted in Eastern China. There is a need to bridge the information gap.

This study aimed to investigate the situation of KAB among the population at risk for hypertension and analyze the impact of socio-demographic factors on the KAB in a district of Shanghai. The results will help us propose interventions to improve the KAB of the high-risk population, which could be of great benefit to delay the onset of hypertension and reduce its prevalence.

## 2. Materials and Methods

### 2.1. Study Design and Sampling

This was a cross-sectional study conducted in 2017. We selected the Jiading District of Shanghai, which is located in the well-developed Eastern Region of China, as the research setting. In 2017, Jiading’s per capita GDP was RMB 104,423 (RMB 1 was about USD 0.148 in 2017), and it ranked the sixth among the 16 districts of Shanghai [20]. Additionally, the research team was in partnership with the Center for Disease Control and Prevention of Jiading for nearly 15 years, and the residents in this district had good compliance with the survey, which could help us better accomplish the survey.

A multi-stage, cluster sampling process was used to select the participants at risk for hypertension. The awareness rate of hypertension-related knowledge was considered as the key indicator to determine the sample size. A sample size of 583 participants was determined based on the confidence level (1 − α = 0.95), allowable error (ε = 0.05) and response rate of the survey (90%). Thus, around 2000 residents in 4 lanes (with approximately 500 to 600 residents in each lane in Jiading) were needed to screen the required high-risk individuals, assuming that the prevalence of prehypertension was about 30% [21]. The sampling was performed in three steps: first, thirteen communities in Jiading were divided into three groups based on their level of economic development: good, medium and poor; second, two communities in the medium group were selected by using a computer-generated random number, and two neighborhood blocks in each sampled community were randomly selected; third, one lane was randomly selected from each sampled neighborhood block. All adult residents in four sampled lanes were invited to participate in our study. Excluding those were absent during screening or not willing to participate in the screening, 1721 residents were screened based on the following screening criteria to reach the required sample size (Figure 1). The inclusion criteria were: adult inhabitants; permanent residence of the community (excluding those who have left this community for more than half year); willing to participate in this study. The exclusion criteria were as follows: those with diagnosed hypertension; those with low cognitive function, unconsciousness, or severe physical illness.

Ethical approval (IRB#2017-TYSQ-03-10) for the study was obtained from the Medical Research Ethics Committee at the School of Public Health of Fudan University, Shanghai, China (IRB00002408 & FWA00002399). All the participants gave written informed consent before data collection.

### 2.2. Data Collection

This study was conducted in August and September 2017. Subjects at risk for hypertension were identified by the trained general practitioners from community health service centers. The criteria were based on Shanghai Guidelines for the Prevention and Treatment of Hypertension in Communities [22]: (1) overweight or obesity (Body mass index (BMI) ≥ 24 kg/m^2^); (2) family history of hypertension; (3) systolic BP between 120–139 mmHg and/or diastolic BP between 80–89 mmHg; (4) long-term excessive drinking (daily intake 100 mL liquor/1200 mL beer/300 mL rice wine/600 mL wine, and more than 4 days per week); (5) long-term high salt diet (daily salt intake ≥ 10 g); (6) having diabetes. Those who met at least one of the criteria were classified as populations at risk for hypertension. BP was measured by the trained general practitioners, using validated Omron electronic sphygmomanometer (OMRON Corporation, Kyoto, Japan). After resting quietly in a seated position for at least 5 min, 3 consecutive BP readings were obtained at 2-min intervals by the automated validated device using appropriate cuff size determined from an upper arm circumference measurement. The height of the upper arm or wrist should be aligned with the heart level. All available readings were used to calculate the mean systolic BP and diastolic BP for each subject. A total of 611 participants at risk for hypertension were included in this study.

A pre-tested, structured questionnaire was used to investigate the participants at risk for hypertension. The questionnaire was conducted face-to-face by researchers at Fudan University and general practitioners. All investigators were trained on the contents of the questionnaire and interview techniques before the investigation. The questionnaire consisted of four sections. Section 1 referred to the socio-demographic characteristics, e.g., gender, age, marital status, educational level, annual income per capita, and medical insurance status. Section 2 referred to the knowledge about hypertension prevention, including 16 items containing basic knowledge about hypertension, risk factors for hypertension, and possible complications. Section 3 referred to the attitude of hypertension prevention, consisting of 10 questions. Section 4 referred to the behavior and lifestyle, including 13 items. Answering options in the latter three sections were dichotomous, i.e., “Know/Don’t Know” or “Yes/No”. The questionnaire showed high internal consistency (Cronbach’s alpha coefficients were 0.838, 0.883, and 0.621 for knowledge, attitude, and behavior sections, respectively).

Completed questionnaires were checked by the research team. In case of any error or missing, the investigators would contact the subject by telephone to correct or add the information.

### 2.3. Variable Definitions

Age was categorized into three groups: high (70 years or above), medium (60–69 years), and low (below 60 years). Educational levels were categorized into three groups: low (primary school or below), medium (junior high school), and high (high school or above). Annual income per capita was categorized into two groups: low (less than RMB 60,000) and high (RMB 60,000 or above). Marital status was categorized into two types: currently married and single. Medical insurance status was categorized into three groups: New Rural Cooperative Medical Insurance (NRCMI), Urban Resident’s Basic Medical Insurance (URBMI), and Urban Employee’s Basic Medical Insurance (UEBMI).

The current smoker was defined as subjects who were smoking and consuming at least one cigarette a day. Current alcohol consumption was defined as subjects who drank alcohol daily or occasionally. Regular consumption of fruits or vegetables was defined as consuming fruits or vegetables more than five times per week [17]. Moderate consumption of meat was defined as consuming meat less than three times per week. Controlling body weight was defined as measuring body weight and calculating BMI at least once every three months [23]. Frequent physical activity was defined as participating in physical activities at least four times per week, and >30 min each time [24]. Regular BP monitoring was defined as measuring BP at least once every three months by themselves or by physicians.

Those who correctly answered more than 11 knowledge questions, more than 7 attitude questions, and more than 9 behavior questions were considered as having good knowledge, having good attitudes, and having good behaviors, respectively.

### 2.4. Data Analysis

Descriptive analysis was performed to show the basic socio-demographic conditions and KAB of the participants. Q-Q Plot and Kolmogorov–Smirnov tests were used to explore the normality of the data. Chi-square tests were used to compare the difference between men and women regarding the KAB of hypertension. Multivariable logistic regression analysis was performed to investigate the associations between the socio-demographic factors and the KAB of the population at risk for hypertension. Six independent variables were included: gender, age (high, medium, low), marital status, annual income per capita (low/high), educational level (low, medium, high), and medical insurance status. *p* values were two-sided, and *p* < 0.05 was considered as statistically significant. All analyses were performed using SPSS 22.0 (SPSS Inc., Chicago, IL, USA).

## 3. Results

### 3.1. Characteristics

The general characteristics of the study population are presented in Table 1. Of all the 611 participants, there were 310 men and 301 women. All the participants were over 30 years old and the mean age was 63.1 years. Approximately 85% of the population had their educational level lower than high school, and 87% had an annual income lower than 60,000 RMB. Most of the population was married. A smaller proportion of the population had NRCMI insurance while about half had URBMI insurance. The mean systolic BP was 130.2 mmHg and the mean diastolic BP was 78.6 mmHg. About half of the participants were overweight.

### 3.2. Hypertension Related KAB

The rates of correct answers regarding knowledge of hypertension ranged between 47.8% and 90.0% (Table 2). The results indicated that approximately half of the participants were not familiar with the diagnostic criteria for hypertension, the Recommended Daily Intake (RDI) of salt, or that heart failure was a complication of hypertension. On the other hand, most of the participants (more than 85%) knew that long-term drinking, high salt intake, and genetic factors were relevant to high BP. It was surprising that about one-fourth of the participants did not know that smoking and low physical activity were related to high BP. There was no difference between males and females regarding knowledge of hypertension.

The rates of positive answers regarding attitudes of hypertension prevention were in general around 80% or higher (Table 3). Among the questions, the risk of smoking on hypertension was of the least awareness (79.2%). There was no difference between males and females regarding attitudes of hypertension prevention.

Approximately three-quarters of the participants did not smoke, or drink alcohol. Most of the participants also had low-risk lifestyles including vegetable consumption, being stress-free, and regular BP monitoring (Table 4). However, about half of the participants did not properly control salt and oil intake, balanced nutrition intake, and control body weight. Only 30.9% of the participants were engaging in frequent physical activities. Compared with males, females had a higher proportion of non-smokers, non-drinkers, and adequate consumption of fresh fruit.

In summary, the study population generally had good knowledge and attitudes of hypertension prevention; however, their behaviors needed great improvement (Table 5).

### 3.3. Factors Associated with One’s KAB

The results from the logistic regressions investigating the socio-demographic factors associated with one’s KAB are presented in Table 6.

The factors significantly associated with knowledge were age and education, where younger participants (Odds Ratio (OR) = 2.82, 95% Confidence Interval (CI): 1.50–5.28 for low age group and OR = 1.64, 95% CI: 1.05–2.54 for medium age group) and participants with higher educational level (OR = 2.24, 95% CI: 1.04–4.84) had better knowledge. The low age group also had better attitudes (OR = 2.87, 95% CI: 1.31–6.26), but not the medium age group (*p* = 0.25). The factors significantly associated with hypertension-related self-management behaviors were being female (OR = 3.05, 95% CI: 2.11–4.41) and UEBMI groups (OR = 2.38, 95% CI: 1.22–4.63). Though age was significant for knowledge and attitudes, it was of little influence on behaviors. The URBMI group also had better behaviors with a borderline significance.

## 4. Discussion

This study provides the latest information on the KAB level of the population at risk for hypertension in the well-developed Eastern Region of China since the implementation of the Essential Public Health Services (EPHS). It indicated that the high-risk population in the sampling area had high knowledge awareness rates and belief formation rates. Since the beginning of the 21st century, both central and local governments in China have put great emphasis on the prevention of key chronic diseases such as hypertension and diabetes, and multiple policies have been released as guidelines [25,26]. In 2009, the national program of EPHS was launched, to provide free essential public health services for all Chinese citizens [27]. The services included the establishment of health records, health education, and chronic disease screening [28]. Many general practitioners were engaged in the prevention and management of chronic diseases. A variety of channels such as TV/broadcast, lectures, columns, and internet were used to promote hypertension prevention knowledge to the residents [28], which probably increased the high-risk group’s knowledge and attitudes. The results of this study confirmed that these strategies have achieved good effects.

There was a noteworthy finding that 85.9% of the participants in this study knew that high salt intake was associated with hypertension, and 84.6% believed that reduced salt intake was helpful to prevent hypertension. However, only 48.4% knew the RDI of salt for adults, which was similar to the results of the studies among hypertensive patients. For example, Gonçalves et al. (2016) [29] observed that 97.4% of the participants knew that high salt intake could lead to hypertension, but only 2.5% knew the RDI of salt in Portugal. This suggested that although many countries had announced the RDI of salt or sodium, the information was not well received by the general public. The scope and strategies of salt reduction/limitation campaign should be further strengthened, including displaying the RDI of salt to residents by salt spoon, informing the residents about the salt reduction knowledge [30], the conversion relationship between condiments and salt, and different kinds of traditional high-salt foods [31]. Additionally, the government should regulate and restrict the amount of salt added in food processing, and promote salt content labels on food packaging [32]. The residents should be educated to check the nutritional facts labels and select processed foods with low sodium [33]. Through all these measures, residents could have a further understanding of the RDI of salt and also the means to control it.

We found that the population participating in UEBMI was more likely to form good health behaviors than those participating in NRCMI, which was similar to the results of Gao et al. [34]. In China, UEBMI mainly covers employees in governmental departments, enterprises, and other institutions, while NRCMI is mainly for farmers or manual workers in rural areas [35]. People participating in UEBMI have more opportunities to be exposed to health education and participate in health promotion activities at their working places, which is beneficial for the formation of healthy behaviors. Employees may be influenced by their colleagues to form a better lifestyle and participate in more physical activities [36]. Furthermore, the working institutions usually organized regular physical examinations to monitor employees’ health status and could find problems in early stages [37]. Therefore, it was easier for people who got UEBMI to acquire and understand the health knowledge and more actively to engage in healthy behaviors.

This study also showed that age, gender, and education level had important influences on hypertension-related knowledge, attitudes, and behaviors, which was consistent with previous studies [13,19]. For instance, the formation of women’s health behaviors was better than men’s. There may be several explanations. First, compared with men, women were more willing to actively take information about health and changes in lifestyle behaviors. For example, the gender difference in fruit and vegetable intake has been widely confirmed [38]. In this study, the proportion of women with adequate consumption of fresh fruit was 48.2%, higher than that of men (35.5%). Second, influenced by Chinese culture, it is more common and acceptable for men to smoke and drink, but not for women [39]. Meanwhile, men may be more exposed to smoking and drinking at social events, especially the working-related ones, which lead to the fact that men smoked and drank more often than women [40]. Therefore, it is necessary to make more targeted interventions to promote the formation of healthy behaviors for men. For example, men should be guided to consume more fruits and vegetables [41]. Male smokers should be educated to restrict their smoking behaviors in public places and at home, or even to quit smoking, by their effort or by going to smoking cessation clinics [42].

Our study has several limitations. First, only six socio-demographic factors were included in the analysis. Unmeasured confounding factors may exist and hamper the results of the study. However, we do not think that the bias induced by unmeasured confounding would be large enough to change our main results and conclusions. Second, we only performed quantitative analysis, yet qualitative interviews could be an alternative way to identify the reasons for the differences. Third, since this study was conducted in a district in Shanghai and the results could be dependent on the local public health education, the conclusions were mainly applicable to the developed regions in Eastern China. Extrapolation to the Central and Western regions should be done with caution. Fourth, health behaviors were self-reported in this study rather than using measuring tools (e.g., International Physical Activity Questionnaire), which could result in reporting bias.

## 5. Conclusions

Our study found that the rates of accurate knowledge and attitudes in the population at risk for hypertension in Eastern China were high, but the formation of healthy behaviors needed further promotion. Groups with older age, lower education, males, and participating in NRCMI had a higher risk for hypertension. Specific public health efforts on further health education and intervention should be made aiming at those groups, which could be of great benefit to delay the onset of hypertension and reduce its prevalence.

## Figures and Tables

**Figure 1 ijerph-17-03683-f001:**
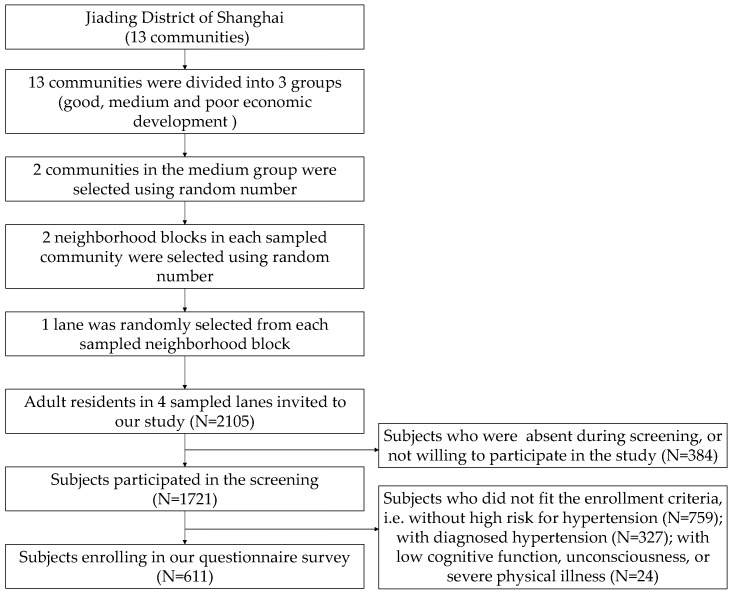
Flow diagram of the sampling and the population enrollment.

**Table 1 ijerph-17-03683-t001:** Characteristics of the study population.

Characteristics	n (%)/Mean ± SD
Gender	
Male	310 (50.7%)
Female	301 (49.3%)
Age [years]	63.1 ± 9.6
Age group	
High (≥70 years)	144 (23.6%)
Medium (60–69 years)	298 (48.8%)
Low (<60 years)	169 (27.6%)
Educational level	
Low (primary school or below)	176 (28.8%)
Medium (junior high school)	340 (55.6%)
High (high school or above)	95 (15.6%)
Annual income per capita	
Low (<60,000 RMB)	533 (87.2%)
High (≥60,000 RMB)	78 (12.8%)
Marriage status	
Single	45 (7.4%)
Married	566 (92.6%)
Medical insurance status	
NRCMI	58 (9.5%)
URBMI	305 (49.9%)
UEBMI	248 (40.6%)
BMI [kg/m^2^]	
Underweight (<18.5)	22 (3.6%)
Normal (18.5–23.9)	243 (39.8%)
Overweight (24–27.9)	277 (45.3%)
Obese (≥28)	69 (11.3%)
SBP [mmHg]	130.2 ± 13.1
DBP [mmHg]	78.6 ± 8.4

NRCMI: New Rural Cooperative Medical Insurance; URBMI: Urban Resident’s Basic Medical Insurance; UEBMI: Urban Employee’s Basic Medical Insurance; BMI: Body Mass Index; SBP: systolic blood pressure; DBP: diastolic blood pressure.

**Table 2 ijerph-17-03683-t002:** Summary of hypertension-related knowledge of participants at risk.

Items	Positive Response [n (%)]			*p* Value
	Total	Male	Female	
Whether hypertension is a life-long disease	495 (81.0%)	252 (81.3%)	243 (80.7%)	0.86
Whether hypertension could be prevented	497 (81.3%)	254 (81.9%)	243 (80.7%)	0.70
Diagnostic criteria for hypertension in adults	339 (55.5%)	174 (56.1%)	165 (54.8%)	0.74
Whether high BP is related to smoking	449 (73.5%)	228 (73.5%)	221 (73.4%)	0.97
Whether high BP is related to long-term drinking	530 (86.7%)	271 (87.4%)	259 (86.0%)	0.62
Whether high BP is related to high salt intake	525 (85.9%)	259 (83.5%)	266 (88.4%)	0.08
Whether high BP is related to overweight or obesity	512 (83.8%)	257 (82.9%)	255 (84.7%)	0.54
Whether high BP is related to low physical activity	458 (75.0%)	231 (74.5%)	227 (75.4%)	0.79
Whether high BP is related to Type A personality	424 (69.4%)	217 (70.0%)	207 (68.8%)	0.74
Whether high BP is related to psychological stress	474 (77.6%)	242 (78.1%)	232 (77.1%)	0.77
Whether high BP is related to lack of sleep	463 (75.8%)	234 (75.5%)	229 (76.1%)	0.86
Whether high BP is related to genetic factors	550 (90.0%)	278 (89.7%)	272 (90.4%)	0.77
Complications of hypertension				
Coronary heart disease	460 (75.3%)	236 (76.1%)	224 (74.4%)	0.62
Stroke	513 (84.0%)	261 (84.2%)	252 (83.7%)	0.87
Heart failure	292 (47.8%)	149 (48.1%)	143 (47.5%)	0.89
RDI of salt for adults	296 (48.5%)	146 (47.1%)	150 (49.8%)	0.49

Notes: *p* values were calculated by the Chi-square tests, showing the difference between males and females for each item. BP: blood pressure; RDI: recommended daily intake.

**Table 3 ijerph-17-03683-t003:** Summary of attitudes of hypertension prevention of participants at risk.

Items	Positive Response [n (%)]			*p* Value
	Total	Male	Female	
People at high risk for hypertension should improve their lifestyle	516 (84.5%)	262 (84.5%)	254 (84.4%)	0.96
Reducing salt intake can help prevent hypertension	517 (84.6%)	257 (82.9%)	260 (86.4%)	0.23
Reducing oil intake can help prevent hypertension	541 (88.5%)	275 (88.7%)	266 (88.4%)	0.89
Quitting smoking can help prevent hypertension	484 (79.2%)	245 (79.0%)	239 (79.4%)	0.91
Reducing alcohol intake can help prevent hypertension	532 (87.1%)	273 (88.1%)	259 (86.0%)	0.45
Controlling body weight can help prevent hypertension	511 (83.6%)	260 (83.9%)	251 (83.4%)	0.87
Exercising can help prevent hypertension	503 (82.3%)	253 (81.6%)	250 (83.1%)	0.64
Reducing psychological stress can help prevent hypertension	505 (82.7%)	257 (82.9%)	248 (82.4%)	0.86
Adequate sleeping can help prevent hypertension	504 (82.5%)	258 (83.2%)	246 (81.7%)	0.62
People at high risk for hypertension should monitor their BP	555 (90.8%)	282 (91.0%)	273 (90.7%)	0.90

Notes: *p* values were calculated by the Chi-square tests, showing the difference between males and females for each item. BP: blood pressure.

**Table 4 ijerph-17-03683-t004:** Summary of hypertension-related behaviors of participants at risk.

Items	Positive Response [n (%)]			*p* Value
	Total	Male	Female	
Not smoking	443 (72.5%)	145 (46.8%)	298 (99.0%)	<0.01
Not drinking alcohol	443 (72.5%)	153 (49.4%)	290 (96.3%)	<0.01
Control of salt intake	274 (44.8%)	134 (43.2%)	140 (46.5%)	0.41
Control of oil intake	307 (50.2%)	154 (49.7%)	153 (50.8%)	0.77
Adequate consumption of fresh fruit	255 (41.7%)	110 (35.5%)	145 (48.2%)	<0.01
Appropriate consumption of meat	362 (59.2%)	162 (52.3%)	200 (66.4%)	<0.01
Adequate consumption of fresh vegetable	572 (93.6%)	292 (94.2%)	280 (93.0%)	0.55
Balanced meat and vegetable intake	233 (38.1%)	132 (42.6%)	101 (33.6%)	0.02
Controlling body weight	298 (48.8%)	142 (45.8%)	156 (51.8%)	0.13
Frequent physical activity	189 (30.9%)	92 (29.7%)	97 (32.2%)	0.49
Low stress from work and life	546 (89.4%)	277 (89.4%)	269 (89.4%)	0.99
Rarely be nervous or panic	519 (84.9%)	279 (90.0%)	240 (79.7%)	<0.01
Monitoring BP regularly	470 (76.9%)	235 (75.8%)	235 (78.1%)	0.50

Notes: *p* values were calculated by the Chi-square tests, showing the difference between males and females for each item. BP: blood pressure.

**Table 5 ijerph-17-03683-t005:** Summary of the proportion of the population that met the criteria for a positive response regarding knowledge, attitudes, and behaviors related to hypertension.

Item Group	Criteria for Positive Response	n (%)
Knowledge	≥11 out of 16	453 (74.1%)
Attitudes	≥7 out of 10	508 (83.1%)
Behaviors	≥9 out of 13	239 (39.1%)

**Table 6 ijerph-17-03683-t006:** Association between the socio-demographic factors and the knowledge, attitudes, and behaviors of the population at risk for hypertension using logistic regression.

Factor		Knowledge		Attitudes		Behaviors	
		OR (95%CI)	*p*	OR (95%CI)	*p*	OR(95%CI)	*p*
Gender	Male *						
	Female	1.15 (0.77–1.71)	0.51	1.28 (0.81–2.04)	0.29	3.05 (2.11–4.41)	<0.01
Age	High (≥70 year) *						
	Medium (60–69 year)	1.64 (1.05–2.54)	0.03	1.34 (0.81–2.21)	0.25	1.05 (0.67–1.63)	0.84
	Low (<60 year)	2.82 (1.50–5.28)	<0.01	2.87 (1.31–6.26)	0.01	0.97 (0.56–1.68)	0.90
Educational level	Low (primary school or below) *						
	Medium (junior high school)	1.38 (0.90–2.12)	0.15	1.11 (0.68–1.83)	0.68	0.93 (0.61–1.41)	0.73
	High (high school or above)	2.24 (1.04–4.84)	0.04	1.87 (0.71–4.91)	0.20	1.37 (0.73–2.58)	0.32
Annual income per capita	Low (<60,000 RMB) *						
	High (≥60,000 RMB)	0.64 (0.33–1.24)	0.18	0.76 (0.34–1.69)	0.50	1.19 (0.67–2.11)	0.55
Marriage status	Single *						
	Married	0.74 (0.36–1.53)	0.41	0.81 (0.35–1.88)	0.63	0.87 (0.46–1.67)	0.68
Medical insurance status	NRCMI *						
	URBMI	0.60 (0.30–1.18)	0.14	0.54 (0.24–1.20)	0.13	1.77 (0.93–3.36)	0.08
	UEBMI	1.00 (0.49–2.07)	0.99	1.28 (0.53–3.09)	0.59	2.38 (1.22–4.63)	0.01

* indicates as reference; OR: odds ratio; CI: confidence interval; NRCMI: New Rural Cooperative Medical Insurance; URBMI: Urban Resident’s Basic Medical Insurance; UEBMI: Urban Employee’s Basic Medical Insurance.

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
