# Peer review of "Hypertension-Related Knowledge, Attitudes, and Behaviors among Community-Dwellers at Risk for High Blood Pressure in Shanghai, China"

_ijerph, 2020, doi:10.3390/ijerph17103683_

Round 1

Reviewer 1 Report

Thank you very much for inviting my review of this interesting manuscript. 

Overall, the manuscript is very well-written.  The abstract and first two paragraphs of the introduction require copy editing to correct English grammar. 

The scientific utility and validity of the work is compelling.  It indicates that implementation of the Essential Public Health Services appears to have had substantial impact on public awareness of cardiovascular risk factors, and that coordination of insured benefits and services, screening, public health education, health promotion activities have potential to rationally target risk reduction strategies to strata of the population that can benefit most, for example to curb smoking and alcohol particularly among working males.  

The study design, sampling design/data collection approach and analysis were methodologically sound.  The inclusion and exclusion criteria were scientifically and ethically appropriate.   The operational definition of pre-hypertension was based on six main contemporary criteria in the Shanghai Guidelines for the Prevention and Treatment of Hypertension [Note:  the citation for these guidelines should be added to the list of references].  The data collection team was deployed with validated blood pressure measurement equipment to follow per guideline standardized blood pressure measurement technique. 

The results of this multi-staged cross-sectional survey logically support the conclusions within the clearly identified limitations for generalization and lacking qualitative basis for attribution of cause to the rates of health-related behaviors. 

Reviewer 2 Report

  1. Methods: sample size calculation is unclear to the reviewer. Which is the main comparison and which is the assumption on the difference? The calculation was also based on the assumption that pre-hypertension would have a prevalence of 30% - this needs to be documented by providing estimates from prior epidemiological studies in the Chinese population.
  2. Methods: The random selection of participants is also unclear – the authors should provide methodological details that reassured the random procedure.
  3. Methods: the study did not include pre-hypertensives; in fact, the study included participants with a variety of established cardiovascular risk factors (i.e. overweight/obese patients, or patients with diabetes mellitus). This issue needs to be clarified.
  4. Methods: the conditions and the exact methodology of BP monitoring are also incomplete – who was responsible for taking the BP measurements? Which method and what type of BP monitor was used?
  5. A study flow diagram of patient evaluation and enrollment is necessary – how many were invited to participate? How many were screened? How many were excluded? The reasons why they were excluded?
  6. Methods: the authors should provide data on the validity of the questionnaire that was used in this survey.
  7. Statistical analysis: which test was used to explore the normality of the distribution of the data?
  8. Statistical analysis: how were these 6 variables were selected for inclusion in multivariable logistic regression analysis? Why only 6 variables? How can the authors address the issue of residual confounding in their analysis?
  9. Table 1: please expand this table in an attempt to improve the description of this cohort – (i.e., basic labs, medications for other conditions, etch).
  10. Results: the overall paper provides only descriptive data separately for men and women, but there is no comparison between men and women.
  11. Results: it would be important to compare the characteristics of participants with prehypertension enrolled in this survey with the characteristics of those with hypertension that were excluded from this study.
  12.  

Round 2

Reviewer 2 Report

I have no other comments ! Thank you very much!